# Comprehensive Profiling of Mutations to Influenza Virus PB2 That Confer Resistance to the Cap-Binding Inhibitor Pimodivir

**DOI:** 10.3390/v13071196

**Published:** 2021-06-22

**Authors:** Y. Q. Shirleen Soh, Keara D. Malone, Rachel T. Eguia, Jesse D. Bloom

**Affiliations:** 1Fred Hutchinson Cancer Research Center, Basic Sciences Division, Seattle, WA 98109, USA; yqsoh@fredhutch.org (Y.Q.S.S.); kmalone2@fredhutch.org (K.D.M.); reguia@fredhutch.org (R.T.E.); 2Fred Hutchinson Cancer Research Center, Computational Biology Program, Seattle, WA 98109, USA; 3Howard Hughes Medical Institute, Seattle, WA 98109, USA

**Keywords:** influenza, drug resistance, pimodivir, PB2, polymerase, deep mutational scanning

## Abstract

Antivirals are used not only in the current treatment of influenza but are also stockpiled as a first line of defense against novel influenza strains for which vaccines have yet to be developed. Identifying drug resistance mutations can guide the clinical deployment of the antiviral and can additionally define the mechanisms of drug action and drug resistance. Pimodivir is a first-in-class inhibitor of the polymerase basic protein 2 (PB2) subunit of the influenza A virus polymerase complex. A number of resistance mutations have previously been identified in treated patients or cell culture. Here, we generate a complete map of the effect of all single-amino-acid mutations to an avian PB2 on resistance to pimodivir. We identified both known and novel resistance mutations not only in the previously implicated cap-binding and mid-link domains, but also in the N-terminal domain. Our complete map of pimodivir resistance thus enables the evaluation of whether new viral strains contain mutations that will confer pimodivir resistance.

## 1. Introduction

Antivirals are an important prophylactic and treatment for influenza, particularly for high-risk patients and those with severe infections [1]. They are also stockpiled as a first line of defense against pandemics caused by novel influenza strains [2], for which vaccines are unlikely to be available for many months.

The first influenza antivirals approved over two decades ago are the adamantanes, which target the matrix 2 (M2) ion channel of influenza A viruses, and neuraminidase inhibitors (NAIs), which inhibit the viral enzyme that facilitates the release of viral progeny from infected cells [3]. Their clinical use, however, has been impacted by the emergence of resistance mutations. Consequently, adamantanes are no longer recommended for clinical use due to resistance in currently circulating influenza strains [4]. NAIs remain the standard of care, though resistance has been shown to emerge in some patients [5], and went to fixation in the now extinct lineage of H1N1 influenza that circulated in humans prior to 2009 [6]. As such, there is an important need for new antivirals that not only extend the clinically beneficial window of treatment but are also effective against adamantane- and NAI-resistant strains.

In recent years, efforts have focused on a new class of antiviral drugs for influenza directed against the influenza polymerase [7,8]. The influenza polymerase is an attractive antiviral target because it plays a pivotal role in the replication and transcription of the viral genome and is highly conserved [9,10]. Baloxavir and favipiravir are the first developed antivirals in this class [11,12]. Baloxivir targets the endonuclease function of the polymerase acidic protein (PA) subunit of the influenza polymerase, thus preventing the transcription of the viral mRNA [13]. Favipiravir is a nucleoside analog that induces chain termination and lethal mutagenesis during viral genome replication [14,15].

Pimodivir (also known as VX-787) is also a member of this new class of antivirals targeting the influenza polymerase complex [16]. Specifically, pimodivir targets the PB2 subunit of the polymerase, preventing its binding to the 7-methyl GTP caps of host-capped mRNAs, thus ultimately inhibiting the first step of viral gene transcription. Importantly, pimodivir is active against a diverse panel of influenza A virus strains, including H5N1 avian influenza [17], highlighting its potential as a first-line treatment against novel pandemic strains.

Identifying resistance mutations can guide the clinical deployment of appropriate antivirals and aid in developing new antivirals that are less susceptible to resistance. Known resistance mutations to pimodivir, as with previous influenza antivirals, have typically been identified one at a time as they arose over the course of antiviral treatment either in patients or in the laboratory in cell culture [16,17,18,19,20,21,22,23]. These mutations are located primarily in the PB2 7-methyl GTP cap-binding pocket [17], as well as the mid-link domain where additional contacts with pimodivir are made [24,25]. However, these mutations likely do not represent the full range of mutations that affect pimodivir resistance.

Here, we fully map pimodivir resistance mutations in high throughput. Specifically, using deep mutational scanning, we quantified how pimodivir resistance was affected by all amino-acid mutations to an avian influenza PB2 that were compatible with viral replication in human cells.

## 2. Materials and Methods

### 2.1. PB2 Mutant Virus Libraries

We previously generated three independent mutant virus libraries containing all single amino-acid-mutations to PB2 from an avian influenza strain [26]. The PB2 mutations were made on a background of reassortant virus using polymerase and nucleoprotein genes (PB2, PB1, PA, NP) from the avian influenza strain A/Green-winged Teal/Ohio/175/1986 (S009) [27], and remaining genes (HA, NA, M, NS) from A/WSN/1933 (H1N1) [28]. These libraries were passaged in A549 cells to select for viruses that encoded all functionally tolerated mutations in PB2 for viral replication in A549 cells. The resulting functional virus was used here for resistance profiling.

### 2.2. Resistance Profiling

To identify resistance mutations, we passaged the mutant virus libraries in A549 cells in the absence or presence of pimodivir, and then identified the mutant viruses that were enriched upon drug selection using deep sequencing. We aimed to passage 1 × 10^6^ TCID50 of each mutant virus library in A549 cells at an MOI of 0.2 in the presence of 50 nM of pimodivir. 4 h prior to infection, we seeded 5 × 10^6^ cells in D10 in a 15 cm dish. Just prior to the infection, we replaced D10 media with WGM with 50 nM of pimodivir. 1 × 10^6^ TCID50 of each virus library was then added to each plate. At 2 h post-infection, we replaced the inoculum with fresh WGM with 50 nM pimodivir. 44 h post-infection, we harvested the viral supernatant. As mock-selected controls, each mutant virus library was also similarly passaged in the absence of pimodivir. Selected and mock-selected viral supernatants were sequenced with a barcoded subamplicon sequencing approach as previously described [26,29].

### 2.3. Analysis of Deep Sequencing Data

Deep mutational scanning sequence data was analyzed using dms_tools2 (https://jbloomlab.github.io/dms_tools2, v. 2.3.0, accessed on 14 April 2021). The differential selection [30] was quantified as the logarithm of the mutation’s enrichment in the pimodivir-selected mutant virus library relative to the mock-selected control library. Sequencing of wildtype DNA plasmid was used as the error control for calculating differential selection.

### 2.4. Data Availability and Source Code

The codes for the analyses are provided as Appendix A and at https://github.com/jbloomlab/PB2_Pimodivir_Resistance (accessed on 4 June 2021). Mean mutation and site differential selection measurements are provided as Appendix A, respectively. Sequencing reads are deposited into the NCBI SRA under BioProject ID PRJNA719471.

### 2.5. Polymerase Activity Assays

Minigenome assays were performed in biological triplicate (starting from independent bacterial clones of each PB2 mutant) in HEK293T cells. We seeded 2.5 × 10^4^ HEK293T cells per well of a 96-well plate. Cells were transfected the next day with 10 ng each of HDM_S009_PB2 (for the respective mutant), HDM_S009_PB1, HDM_S009_PA, HDM_S009_NP, 30 ng of pHH-PB1-flank-eGFP reporter, and 30 ng of pcDNA-mCherry as the transfection control using BioT. At 22 h post-transfection, cells were trypsinized and analyzed by flow cytometry. We report minigenome activity as the percent of mCherry-positive cells that are GFP-positive.

### 2.6. Analysis of PB2 Sequence Variation

Sequences of recent influenza strains were downloaded from the Influenza Virus Resource Database in June 2021. Full coding sequences were obtained for human H3N2 seasonal influenza (sequences collected from 2015–2021), human pandemic H1N1 influenza (2015–2021), all subtypes of avian influenza (2015–2021), and human infections of H5N1 and H7N9 avian influenza strains (all sequences available). Unique sequences were aligned using phydms_prepalignment (http://jbloomlab.github.io/phydms/phydms_prepalignment.html, accessed on 14 April 2021). The sequences, alignments, and analysis are available at https://github.com/jbloomlab/PB2_Pimodivir_Resistance (accessed on 4 June 2021).

## 3. Results

In a prior study, we generated triplicate mutant virus libraries containing all single-amino-acid mutations to PB2 from an avian influenza strain, S009, with the other polymerase complex genes (PB1, PA, and NP) also derived from S009, and the remaining genes derived from the lab-adapted A/WSN/1933 (H1N1) strain [26]. We passaged each library to obtain viruses capable of replication in the A549 human lung epithelial carcinoma cell line [26]. In the present study, we passaged these mutant virus libraries once more in A549 cells in the presence or absence of 50 nM of pimodivir, a concentration chosen so that only a small fraction (1–10%) of viral titers was recovered compared to a mock selection (Appendix A). To quantify the effect of each mutation on viral growth in the presence of pimodivir, we sequenced the passaged viruses and then calculated the enrichment of each mutation in the pimodivir-selected vs. nonselected conditions. This value, termed hereafter the “differential selection”, reflects how favorable a mutation is in the presence over the absence of pimodivir.

Selection of pimodivir resistance mutations was highly reproducible across three biological replicates (Appendix A). In addition to previously identified resistance mutations, we identified many more mutations, both at sites at which mutations were previously found, as well as new sites (Figure 1 and Appendix A). Many of these mutations are evolutionarily accessible by single nucleotide substitutions from currently circulating PB2 sequences (Figure 1B–E). Resistance mutations occur in three regions of the PB2 protein.

The first region is the 7-methyl GTP cap-binding pocket (Figure 1B, Figure 2A and Appendix A). Resistance mutations in this region, in combination with structural studies, provide support for the proposed mechanism that pimodivir occupies this pocket and interacts with PB2 similarly to the m^7^GTP guanine base [17]. Resistance mutations previously identified in this region are located at sites K376, which forms hydrogen bonds to both m^7^GTP and pimodivir, H357, F404, and F323, which form aromatic side chain interactions with the azaindole and pyrimidine rings of pimodivir, Q406, which interacts with the carboxylic acid of pimodivir, as well as sites S337, F363, and T378. For each of these sites with known resistance mutations, we identify additional substitutions at each of these sites that also confer resistance, e.g., sites 337 and 376. Sites that appear more tolerant of mutations, e.g., H357, have a correspondingly larger number of resistance mutations at that site, and vice versa, e.g., F404 (Figure 1B).

The second region is the mid-link domain (Figure 1C, Figure 2B,C and Appendix A). Previous studies had identified resistance mutations, such as N510T [17], that lie outside the cap-binding pocket. A more recent structural characterization of pimodivir in complex with the PB2 protein cap-binding and mid-link domain revealed that in the presence of pimodivir, the polymerase can also take up the transcriptionally inactive “apo” configuration. In this configuration, the mid-link and cap-binding domain are stabilized by interdomain interactions, as well as by contacts between pimodivir and the mid-link domain [24,25]. Pimodivir binding is thus proposed to have a second indirect mode of action that is the stabilization of this transcriptionally inactive “apo” state, thus occluding this cap-binding pocket. In support of this model, we identify numerous mutations at sites postulated to play a role in stabilizing the inhibitor bound “apo” state, such as at site N510. We also identify resistance mutations at sites such as S514, which do not appear to directly contact pimodivir but are in proximity to other segments of the mid-link domain. The substitution of serine by various bulky side-chains, such as arginine and glutamine, may destabilize the interaction between the cap-binding and mid-link domain, countering the stabilizing effect of pimodivir.

Unexpectedly, we identified resistance mutations in the PB2 N-terminal domain, such as at sites E188, E192, D195, and C196, located far away from where pimodivir binds (Figure 1D, Figure 2D and Appendix A). Strikingly, E188 and E195 are located side by side on an outward-facing side of an alpha-helix, and D195 and C196 are on an adjacent loop that is also on the surface of the PB2 protein. The most selected amino acids at these sites in the presence of pimodivir are basic amino acids including arginine, histidine, and lysine. Further investigation will be needed to fully understand the role of these sites in pimodivir resistance. Additionally, we also identified sites of high positive differential selection at sites known to be important in human adaptation, namely 627 and 701 (Figure 1E). Thus, it appears that in the context of an avian influenza PB2, human adaptive mutations confer an advantage for influenza replication in human cells in the presence of pimodivir.

To validate that our high-throughput approach accurately identified mutations that countered the inhibitory effects of pimodivir on polymerase activity, we quantified the effect of mutations in a minigenome assay in the presence of a range of pimodivir concentrations (Figure 3A–C). We individually validated a selection of mutations from each of the three regions described above. A few new mutations were selected from two sites at which prior resistance mutations had been identified (F323L, H357D, M431E), and the remaining mutations were in sites that had not yet been associated with pimodivir resistance. Each mutation was made on the background of PB2 from the S009 avian influenza strain that had the E627K human-adaptive mutation. All mutations selected for validation increased pimodivir resistance, raising the EC_50_ between 2 to almost 30-fold (Figure 3E). Some of the resistance mutations, particularly those in the mid-link domain, appear to be detrimental to minigenome activity (Figure 3D). Hence, there may exist a trade-off between resistance to pimodivir and baseline polymerase activity.

Finally, we asked whether PB2 resistance mutations, especially those new ones identified here, may already exist in circulating influenza strains. We examined PB2 sequences from all subtypes of avian influenza, human-infecting avian influenza (H5N1 and H7N9), as well as seasonal and pandemic influenza (H3N2 and H1N1), looking closely at sites associated with pimodivir resistance (Figure 4). Consistent with previous reports, we found that PB2 was very well-conserved. Pimodivir resistance mutations are not observed at significant frequencies in any of the strains analyzed. Hence, we expect pimodivir to be effective against these examined strains. We note, however, that many pimodivir resistance mutations are evolutionarily accessible by single nucleotide substitutions from currently circulating PB2 sequences. Therefore, surveillance efforts are critical for the early identification of any resistant influenza variants as they arise.

## 4. Discussion

We have quantified how pimodivir resistance is affected by all single-amino-acid mutations to an avian influenza PB2 that are compatible with viral replication in human cells. Our comprehensive mapping identified not only many previously known resistance mutations but also new mutations at previously identified sites and new sites. The identified mutations in the cap-binding and mid-link regions of the PB2 protein lend support to current proposed mechanisms of drug action and thus resistance, namely that of the direct inhibition of capped-RNA binding [17], as well as the indirect stabilization of the transcriptionally inactive apo configuration of the cap-binding and mid-link domains [24,25].

A third set of mutations in the PB2 N-terminal domain is unexplained by existing models of pimodivir action. Their characteristics suggest that they may interact with other domains or subunits of the polymerase or other proteins—based on existing structures, they do not contact pimodivir directly. The sites of these mutations (E188, E192, D195, C196) are in a tight cluster on the surface of the protein. Finally, basic amino acids appear to be favored across these sites.

In our current study, we have thus identified numerous novel pimodivir resistance mutations and validated several of them by a minigenome polymerase assay. Further studies, including the validation and characterization of these resistance mutations in a recombinant virus, will be required to understand how these mutations confer pimodivir resistance.

Finally, our comprehensive map of resistance mutations provides empirical data with which we can evaluate new viral strains for pimodivir resistance, including those mutations that would not have been predicted by structural analyses. This in turn will enable the rapid determination of the best antiviral approach to be clinically deployed.

We began this work while pimodivir was in active development and in the midst of Phase 3 clinical trials. Recently, in September 2020, the development of pimodivir for the treatment of influenza was halted due to interim analyses of Phase 3 data that showed that pimodivir, while effective, did not demonstrate an added benefit to the current standard of care (https://www.janssen.com/janssen-discontinue-pimodivir-influenza-development-program, accessed on 4 June 2021). Our pimodivir resistance map, while not immediately applicable at the moment to an approved antiviral therapy, is nevertheless useful in the case that resistance against existing antivirals leads us to revisit pimodivir as an alternate antiviral. Further, such maps of antiviral resistance may help us design the next generation of PB2 inhibitors that are less susceptible to the evolution of resistance. More generally, we demonstrate the utility of deep mutational scanning in evaluating drug resistance and thus facilitating informed clinical responses during pandemic outbreaks.

## Figures and Tables

**Figure 1 viruses-13-01196-f001:**
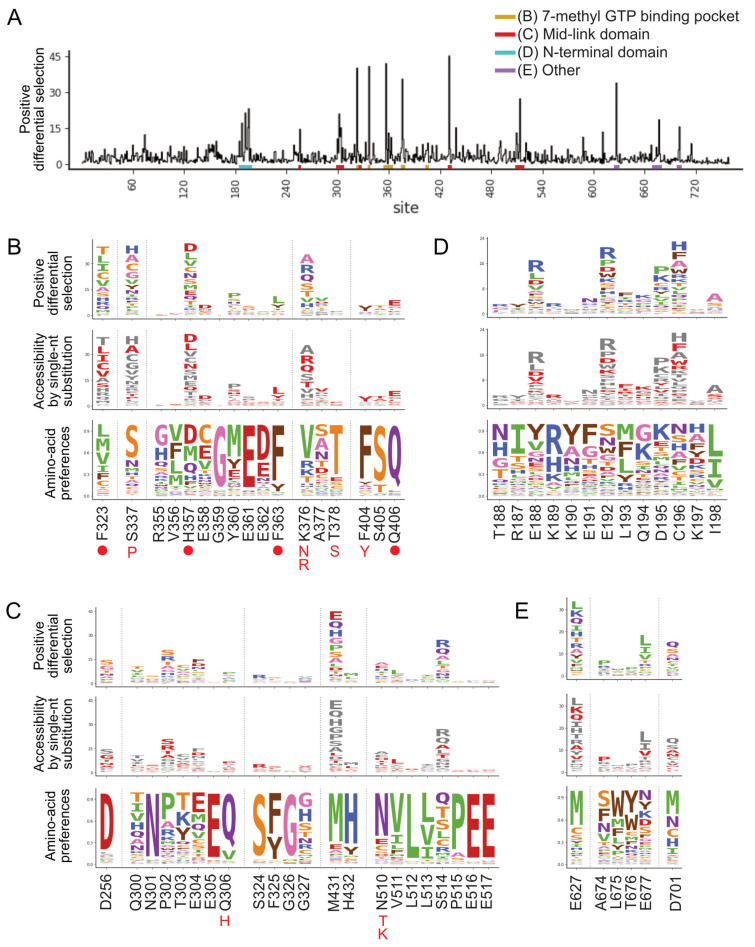
A complete map of pimodivir resistance. (**A**) Positive site differential selection across the entire PB2 protein. Regions of interest from different domains are underlined in different colors. (**B**–**E**) Mutation-level profiles for regions of interest in the (**B**) 7-methyl GTP binding pocket, (**C**) mid-link domain, (**D**) N-terminal domain, (**E**) other regions of PB2. The top plot of each panel shows the positive differential selection profile, where the height of each amino acid is proportional to its differential selection. The middle plot shows which amino acids are accessible by single-nucleotide substitution from existing avian PB2 sequences. Accessible amino acids are shown in red. The bottom plot shows the amino acid preferences at each site in human A549 cells, as previously measured in [26]. The preference for an amino acid is proportional to its enrichment during the prior functional selection of the complete PB2 mutant library in A549 cells. The height of each letter is proportional to the preference for that amino acid at that site. Sites of known resistance mutations are indicated by either a circle underneath the site or the specific resistance mutation if it is known.

**Figure 2 viruses-13-01196-f002:**
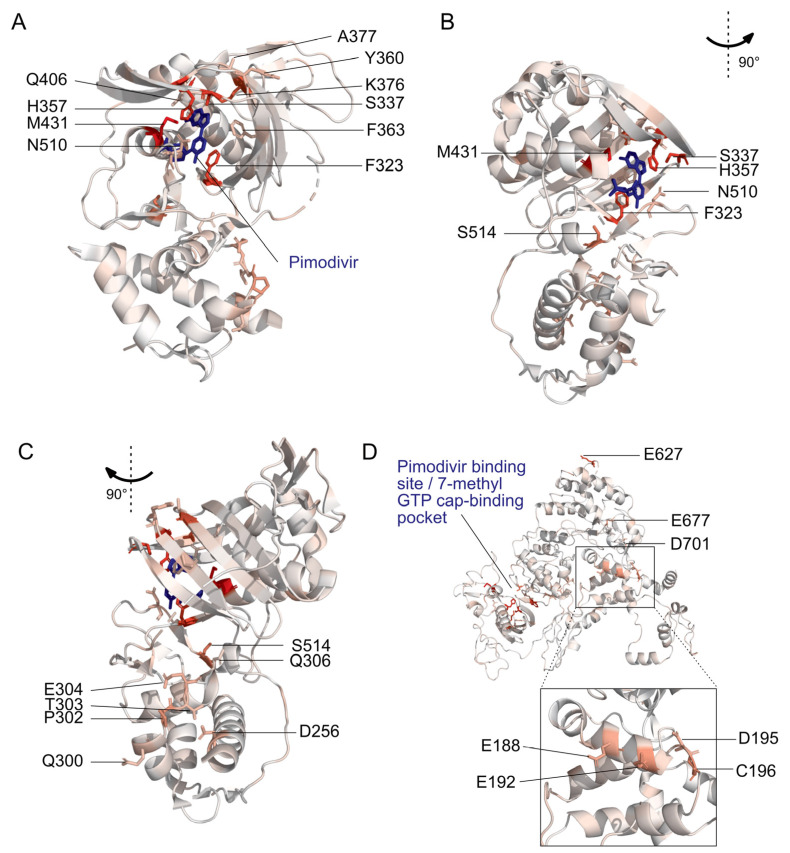
Locations of pimodivir resistance mutations on the PB2 protein. Amino acids are colored on a scale of white to red according to the total positive differential selection at that site, with red corresponding to a high differential selection. Sites of interest, as identified in Figure 1, are labeled. (**A**–**C**) Structural model of PB2 cap-mid-link double domain with pimodivir in the transcriptionally inactive “apo” configuration [25]. Sites with resistance mutations that are in the 7-methyl GTP binding pocket and mid-link domain are labeled here. PDB: 6EUV. (**D**) Structural model of full length PB2 in the apo configuration, but without pimodivir [31]. Sites of resistance mutations that are in the N-terminal domain and elsewhere are labeled here. PDB: 5D98. Appendix A accompanies this figure.

**Figure 3 viruses-13-01196-f003:**
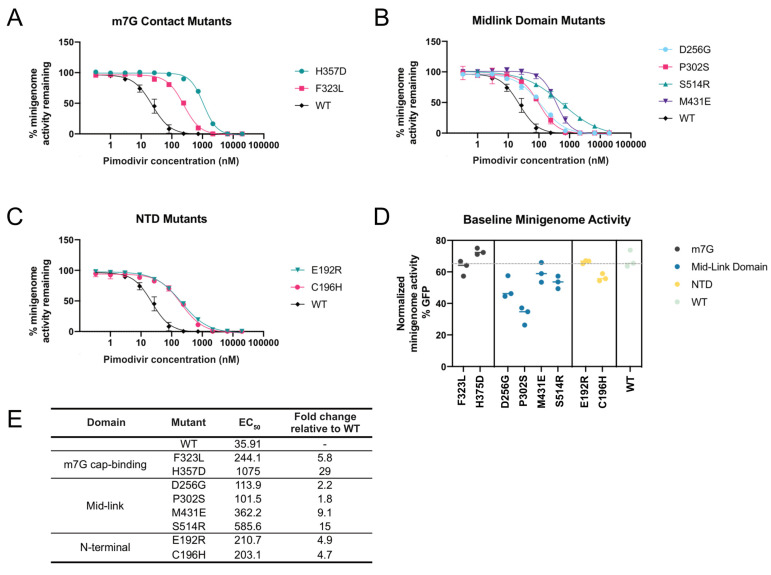
Validation of pimodivir resistance mutations using a minigenome assay. The indicated mutations were all made in the background of PB2-S009 with the E627K human-adaptive mutation. Hence, the “wildtype control”, “WT”, refers to PB2 that already has the E627K mutation. (**A**–**C**) Minigenome activity curves were plotted based on minigenome activity at various pimodivir concentrations, normalized to activity in the absence of pimodivir. (**D**) Minigenome activity was measured for polymerases containing each of the mutations, in the absence of pimodivir. (**E**) EC_50_ and fold-change in EC_50_ relative to wildtype, determined from the fit four-parameter logistic curves.

**Figure 4 viruses-13-01196-f004:**
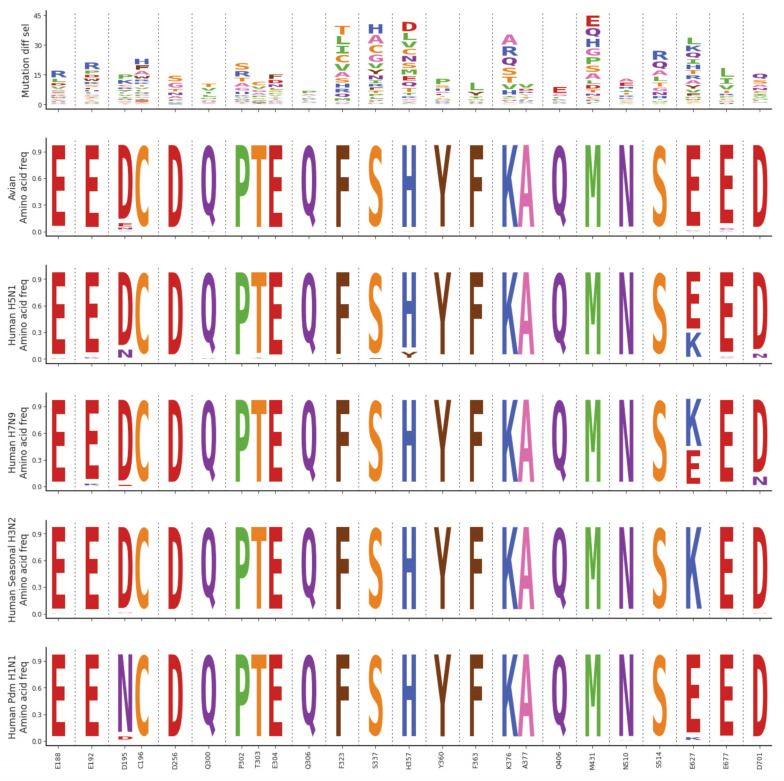
Natural sequence variation at sites of pimodivir resistance. Logoplots representing mutation differential selection under pimodivir selection (top row) and amino acid frequencies in various influenza strains (remaining rows). For the plot representing the mutation differential selection, the height of each amino acid is proportional to its differential selection. For the plots representing the amino acid frequency, the height of each amino acid is proportional to its frequency observed in the specified influenza strains.

## Data Availability

Code for the analyses is provided as Appendix A and at https://github.com/jbloomlab/PB2_Pimodivir_Resistance (accessed on 4 June 2021). Mean mutation and site differential selection measurements are provided as Appendix A, respectively. Sequencing reads are deposited into the NCBI SRA under BioProject ID PRJNA719471.

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
