# Peer review of "Comprehensive Profiling of Mutations to Influenza Virus PB2 That Confer Resistance to the Cap-Binding Inhibitor Pimodivir"

_viruses, 2021, doi:10.3390/v13071196_

Round 1
Reviewer 1 Report
Major concern:
- Pimodivir drug sensitivity of PB2 mutants especially the ones that locate in the N-terminal domain should be further characterized. Reverse genetics to generate the mutant viruses and test pimodivir antiviral activity against the recombinant viruses that contain the PB2 mutations.
Major concern:
- Baloxavir as an approved antiviral for influenza should be included in paragraph 2;
- some typo need to be corrected: EC50 should be EC50; 50nM should be 50 nM;
- Figure 3D has mismatched information.
Author Response
We thank Reviewer 1 for their helpful suggestions.
Major concern:
1. Pimodivir drug sensitivity of PB2 mutants especially the ones that locate in the N-terminal domain should be further characterized. Reverse genetics to generate the mutant viruses and test pimodivir antiviral activity against the recombinant viruses that contain the PB2 mutations.
We agree that the further characterization of the mechanisms of putative resistance mutations, including the ones in the N-terminal domain, is an interesting area of future work. However, the current scope of our study is limited to deep mutational scanning on live virus to identify resistance mutations, followed by validation in mini-genome polymerase assay activity. Generating specific mutations in live virus and characterizing their mechanisms is beyond the scope of our study. We have added text to the Discussion explicitly noting that we have not characterized these mutations beyond the initial deep mutational scan in live virus and subsequent mini-genome assays, and that this is an important area for future work.
Major concern:
1. Baloxavir as an approved antiviral for influenza should be included in paragraph 2;
We have added a brief introduction to baloxavir following paragraph
2. some typo need to be corrected: EC50 should be EC50; 50nM should be 50 nM;
We have made the appropriate typographical corrections.
3. Figure 3D has mismatched information.
We have corrected the figure labels.
Reviewer 2 Report
This study is about drug resistant mutations of a inhibitor Pimodivir (VX-787) to influenza PB2 protein. By using comprehensive mapping with PB2 mutant virus libraries of A/Green-winged Teal/Ohio/175/1986 (H2N1), many previously known resistance mutations and also new mutations were identified in the cap-binding and mid-link regions of PB2 protein, and unexpectedly also in the PB2 N-terminal domain. The results are sound and clearly presented.
Some minor points:
- What about the sequence conservation of the PB2 protein in this study with modern day influenza strains such as the seasonal flu PB2 proteins and some other human-infecting avian virus PB2 proteins such as H5N1, H7N9 etc. A sequence alignment with the drug resistance mutations labeled would be good.
- Figure 2, for highlighted amino acid residues would be more clear to just show side-chains, not the main-chains. The residue labels are too close. The amino acid coloring scale from white to red is hard to tell. As the positive differential selection is already shown in Figure 1, it's not necessary to show it here again.
Author Response
This study is about drug resistant mutations of a inhibitor Pimodivir (VX-787) to influenza PB2 protein. By using comprehensive mapping with PB2 mutant virus libraries of A/Green-winged Teal/Ohio/175/1986 (H2N1), many previously known resistance mutations and also new mutations were identified in the cap-binding and mid-link regions of PB2 protein, and unexpectedly also in the PB2 N-terminal domain. The results are sound and clearly presented.
We thank Reviewer 2 for their positive feedback and helpful suggestions.
Some minor points:
1. What about the sequence conservation of the PB2 protein in this study with modern day influenza strains such as the seasonal flu PB2 proteins and some other human-infecting avian virus PB2 proteins such as H5N1, H7N9 etc. A sequence alignment with the drug resistance mutations labeled would be good.
We have added a new figure (Figure 4) showing an analysis of sequence conservation of PB2 in seasonal and pandemic influenza (H3N2 and H1N1), human-infecting avian influenza (H5N1 and H7N9), and avian influenza (all subtypes). Briefly, PB2 is very well conserved, and pimodivir resistance mutations are not observed at significant frequencies in any of the strains analyzed.
2. Figure 2, for highlighted amino acid residues would be more clear to just show side-chains, not the main-chains. The residue labels are too close. The amino acid coloring scale from white to red is hard to tell. As the positive differential selection is already shown in Figure 1, it's not necessary to show it here again.
We simplified the main figure by reducing the number of residues with side chains labeled to the ones with the highest differential selection. We would prefer to retain the white-red coloring of the main chain based on positive differential selection, as it shows the reader the location of these sites on the protein – this spatial information is not represented in Figure 1. However, we have also made a new accompanying supplemental Figure S3 based on Reviewer 2 suggestions. In this further simplified figure, we show the same structures, coloring only the side chains of the selected residues, and not the main chains.
Reviewer 3 Report
This manuscript by Soh and co-workers reports a study concerning different mutations of polymerase basic protein 2 (PB2) subunit of the influenza A virus polymerase complex capable of conferring resistance to the cap-binding inhibitor pimodivir. In particular, a complete map of all single amino acid mutations of PB2 subunit of an avian influenza A virus polymerase responsible for resistance to pimodivir has been generated. Resistance mutations have been observed in three regions of the PB2 protein.
Both known and new resistance mutations have been identified and it is expected that the complete map of pimodivir resistance will allow the evaluation of whether the new viral strains contain mutations that confer resistance to pimodivir.
The manuscript is interesting, well written and understandable to a specialist readership. The organization and structure of the paper are satisfactory.
The title clearly indicates the focus of the study and the abstract section well summarizes the article contents. The introduction provides sufficient background and the objective of the study is clearly stated. "Materials and Methods" are appropriate and contain information understandable to a reader particularly skilled in the subject. The results are exhaustively analysed and Figures (as well Supplementary files) are well designed and all necessary for understanding of the text.
The conclusion provides both the critical interpretation of the results and the implications for future research.
The identification of new the mutations responsible for drug resistance is of particular importance as, by defining the mechanisms of resistance, it can guide the development of new antivirals.
I have just a few suggestions.
The "References" section must be completely revised. Most of the references are incomplete (lack of volume number, page numbers .….).
In particular, in the referee opinion, the references should be increased in number.
For example, it is important to mention at least these papers:
Gregor J, Radilová K, Brynda J, Fanfrlík J, Konvalinka J, Kožíšek M. Structural and Thermodynamic Analysis of the Resistance Development to Pimodivir (VX-787), the Clinical Inhibitor of Cap Binding to PB2 Subunit of Influenza A Polymerase. Molecules. 2021 Feb 14;26(4):1007. doi: 10.3390/molecules26041007. PMID: 33673017
Mengual-Chuliá B, Alonso-Cordero A, Cano L, Mosquera MDM, de Molina P, Vendrell R, Reyes-Prieto M, Jané M, Torner N, Martínez AI, Vila J, Díez-Domingo J, Marcos MÁ, López-Labrador FX. Whole-Genome Analysis Surveillance of Influenza A Virus Resistance to Polymerase Complex Inhibitors in Eastern Spain from 2016 to 2019. Antimicrob Agents Chemother. 2021 May 18;65(6):e02718-20. doi: 10.1128/AAC.02718-20. Print 2021 May 18. PMID: 33782005
Takashita E. Influenza Polymerase Inhibitors: Mechanisms of Action and Resistance. Cold Spring Harb Perspect Med. 2021 May 3;11(5):a038687. doi: 10.1101/cshperspect.a038687. PMID: 32122918; PMCID: PMC8091960.
Hayden FG, Shindo N. Influenza virus polymerase inhibitors in clinical development. Curr Opin Infect Dis. 2019 Apr;32(2):176-186. doi: 10.1097/QCO.0000000000000532. PMID: 30724789; PMCID: PMC6416007.
Author Response
This manuscript by Soh and co-workers reports a study concerning different mutations of polymerase basic protein 2 (PB2) subunit of the influenza A virus polymerase complex capable of conferring resistance to the cap-binding inhibitor pimodivir. In particular, a complete map of all single amino acid mutations of PB2 subunit of an avian influenza A virus polymerase responsible for resistance to pimodivir has been generated. Resistance mutations have been observed in three regions of the PB2 protein.
Both known and new resistance mutations have been identified and it is expected that the complete map of pimodivir resistance will allow the evaluation of whether the new viral strains contain mutations that confer resistance to pimodivir.
The manuscript is interesting, well written and understandable to a specialist readership. The organization and structure of the paper are satisfactory.
The title clearly indicates the focus of the study and the abstract section well summarizes the article contents. The introduction provides sufficient background and the objective of the study is clearly stated. "Materials and Methods" are appropriate and contain information understandable to a reader particularly skilled in the subject. The results are exhaustively analysed and Figures (as well Supplementary files) are well designed and all necessary for understanding of the text.
The conclusion provides both the critical interpretation of the results and the implications for future research.
The identification of new the mutations responsible for drug resistance is of particular importance as, by defining the mechanisms of resistance, it can guide the development of new antivirals.
We thank Reviewer 3 for their positive feedback and helpful suggestions.
I have just a few suggestions.
The "References" section must be completely revised. Most of the references are incomplete (lack of volume number, page numbers .….).
We have inspected and revised the References to ensure they are complete.
In particular, in the referee opinion, the references should be increased in number.
We have now included these suggested references, as well as others added to response to other reviewer comments.
For example, it is important to mention at least these papers:
Gregor J, Radilová K, Brynda J, Fanfrlík J, Konvalinka J, Kožíšek M. Structural and Thermodynamic Analysis of the Resistance Development to Pimodivir (VX-787), the Clinical Inhibitor of Cap Binding to PB2 Subunit of Influenza A Polymerase. Molecules. 2021 Feb 14;26(4):1007. doi: 10.3390/molecules26041007. PMID: 33673017
Mengual-Chuliá B, Alonso-Cordero A, Cano L, Mosquera MDM, de Molina P, Vendrell R, Reyes-Prieto M, Jané M, Torner N, Martínez AI, Vila J, Díez-Domingo J, Marcos MÁ, López-Labrador FX. Whole-Genome Analysis Surveillance of Influenza A Virus Resistance to Polymerase Complex Inhibitors in Eastern Spain from 2016 to 2019. Antimicrob Agents Chemother. 2021 May 18;65(6):e02718-20. doi: 10.1128/AAC.02718-20. Print 2021 May 18. PMID: 33782005
Takashita E. Influenza Polymerase Inhibitors: Mechanisms of Action and Resistance. Cold Spring Harb Perspect Med. 2021 May 3;11(5):a038687. doi: 10.1101/cshperspect.a038687. PMID: 32122918; PMCID: PMC8091960.
Hayden FG, Shindo N. Influenza virus polymerase inhibitors in clinical development. Curr Opin Infect Dis. 2019 Apr;32(2):176-186. doi: 10.1097/QCO.0000000000000532. PMID: 30724789; PMCID: PMC6416007.